# Addressing bias in national population density models: Focusing on rural Senegal

**Corentin Visée** [1,2] *, **Camille Morlighem** [1,2], **Catherine Linard** [1,2], **Abdoulaye Faty** [3], **Sabine Henry** [1,2], **Sébastien Dujardin** [1]

1 Department of Geography, Faculty of Sciences, University of Namur, Namur, Belgium, 2 Institute of Life, Earth, and Environment, University of Namur, Namur, Belgium, 3 Department of Geography, University of Cheikh Anta Diop, Dakar, Senegal

* corentin.visee@unamur.be

**Data Availability Statement:** The data and codes that support the findings of this study are available with the identifier at the link: https://figshare.com/s/dc36e360ae0391a1d2a8.

## Abstract

Knowing where people are is crucial for policymakers, particularly for the efficient allocation of resources in their country and the development of effective, people-centred policies. However, rural population distribution maps suffer from biases related to the type of dataset used to predict population density, such as the use of nighttime lights datasets in areas without electricity. This renders widely used datasets irrelevant in rural areas and biases nationwide models towards urban areas. To compensate for such biases, we aim at understanding the importance and relationship between water-related covariates and population densities in a random forest model across the urban-rural gradient. By extending a recursive feature elimination framework, we show that commonly used covariates are only selected when modelling the whole country. However, once the highest density areas are removed, water-related characteristics (especially distance to boreholes) become important covariates of population density outside of densely populated areas. This has important implications for modelling population in rural areas, including for a better estimation of the size of remote communities. When seeking to produce country-level population maps, we encourage further studies to explicitly account for rural areas by considering the urban-rural gradient and encourage the use of water-related datasets.

## Introduction

Policymakers need accurate and up-to-date maps of population density for their country to allocate resources efficiently in the event of disasters or to develop effective policies [1–4]. In addition, global assessment of the impact of climate change on populations requires a global, easy-to-use and reproducible methodology to understand the consequence of disasters on populations [5, 6]. Early studies on this topic used censuses, considered the gold standard, as the main source of information on the location of populations [7]. They provide a comprehensive source of population by enumeration units (e.g., administrative divisions) within a country [8]. However, censuses are not time-synchronised between countries, the spatial aggregation of census data is country-specific and rural areas are not always explicitly mentioned. It is

**Funding:** The authors received no specific funding for this work.

**Competing interests:** The authors have declared that no competing interests exist.

therefore complex to produce accurate and comparable maps of population density across countries and over time, as relationship between population distribution and environmental and socio-economic characteristics can be highly context dependent [9].

Well-known approaches include initiatives such as WorldPop, LandScan or the United Nations Environment Programme. The aim of these initiatives is to link census data with external datasets using machine learning methods [10]. [11] extended the areal weighting scheme first introduced by [12] by linking remote sensing and open-source georeferenced data in a random forest (RF) model. This approach exploits the information existing between selected datasets (e.g., elevation, land cover, location of urban settlements or nighttime light) and census data to redistribute population within administrative units at a given spatial resolution [1, 13]. This approach uses a unique model to learn potential relationships between population density and selected datasets in rural and urban administrative units within countries. Such a type of a national model used to link population density with socio-economic and environmental for the whole country does not mention explicitly rural and urban areas. Indeed, there are high disparities in the access to electricity [14] or in the economic activities carried out [15], which affects the representativity of the country-wide model in rural areas. For instance, the lack of electricity in remote areas makes nighttime light an unreliable covariate of population density, while it is considered as an important covariate of density in most country-wide model built in the literature [13, 16, 17]. In addition, the quality of crowd-sourced datasets such as OpenStreetMap, Twitter, Facebook and Tencent varies considerably between urban and rural contexts [18–21]. As a result, while the relative importance of data sets commonly used varies within countries and along their urban-rural gradient, no studies quantified this variation in importance.

In the rural setting, water fetching activities are of paramount importance for rural communities in Sub-Saharan Africa. Fetching water is time and energy consuming for people in rural areas, who have to walk and wait at the water point [22, 23]. There is also a high risk of injury during the journey due to road quality and insecurity [24]. Furthermore, the urban-rural divide affects the water sources used, as the choice of water source can be influenced by wealth, and cities are often connected by piped water, reducing the burden of water collection but making the boreholes, wells and surface water the primary source of water in rural areas [25, 26]. As rural areas are often the poorest and least prepared to cope with environmental change, they are also the most vulnerable to natural disasters and environmental changes such as water scarcity and drought [1]. This vulnerability to natural disasters is exacerbated by the poor understanding of the spatial distribution of rural populations, enhancing the need for studies on the drivers of rural population distribution within countries [1]. As population estimates based on remote sensing and crowd-sourced data can be unreliable in rural areas, there is a need for other types of data sources [27, 28].

In this paper, we aim to understand the mains factors influencing population density along an urban-rural gradient in order to avoid the under-representation of rural areas, by adding covariates representative of these areas such as water-related features and comparing their importance to common drivers of population densities. To assess the relative importance of covariates, we innovate by building the random forest models within a recursive feature elimination framework that account for spatial autocorrelation. To reach consensus on important covariates, we extended the recursive feature elimination by making multiple iteration. Given the large rural-urban disparities in Senegal and the complex relationship between environment, socio-economic context, and population density, we tested four separate models for four different population density batches.

## Materials and methods

### Materials

Covariates used in the population density modelling include average nighttime light data, land cover and land use or economic facilities, such as railways and roads. We have added climatic covariates as the Normalised Difference of Vegetation Index (NDVI) and other vegetation indices. An overview of dataset used is given in Table 1. For this study, we added a set of 14 water-related covariates, which we describe below (see Table 1).

**Accessibility & socio-economic infrastructure.** *Boreholes and water tower*. Senegal has a total of 4302 boreholes and 163 water towers distributed throughout over the country in 2014 (Fig 1). To understand the link between water availability infrastructure and population density, the geographical distance to boreholes and water towers in Senegal is calculated for the whole country. Overall, each administrative level has between 0 and 70 boreholes, with a mean number of 9.90 and a standard deviation of 11.07. In Senegal, all new boreholes made to fetch water (for personal or agricultural use) were registered by the *Direction de la Gestion et de la Planification des Ressources en Eau* (DGPRE). Although it may affect the use of water for domestic and agricultural purposes, we did not consider water quality in this study and assumed that all water points provided water of the same consistency and quality over the period considered.

*River network and coastline*. Senegal has two major river basins: the Senegal Basin in the northern part of the country and the Gambia River, which also drains the Gambia, next to Senegal. As water availability is partly determined by surface water [29], the distance to the river network is calculated for the whole of Senegal.

**Table 1. Covariates and datasets used in this study.** Water-related datasets are highlighted in bold and underlined.

| Category | Covariates | Spatial resolution (m) | Year | Source |
|---|---|---|---|---|
| *Accessibility & socio-economic infrastructure* | **Distance to boreholes and water tower** | 1000 | 1931–2014 | DGPRE |
| | Distance to riverways, roads, railways, education facilities, airports, points of interest, financial services, and health site. | 1000 | Extracted in June 2023 | Humanitarian Data Exchange (HDX) |
| | Distance to points of interest, roads, railways, riverways, traffic facilities, transport facilities, populated places and religious points | 1000 | Extracted in June 2023 | OpenStreetMap (OSM) |
| | Distance to cities | 1000 | Extracted in June 2023 | World Cities Database |
| | **NDMI**, **NDWI**, NDVI, NPP | 10 | 2015–2017 | Sentinel-2 |
| | Nighttime light | 500 | 2016 | Earth Observation Group |
| *Climate & environment* | Annual mean temperature, **annual mean precipitation**, mean temperature of wettest and driest quarter, **mean precipitation**, mean solar radiance, mean temperature, max temperature | 1000 | 1970–2000 | WorldClim |
| | Mean temperature of warmest and coldest quarter, **precipitation of driest and wettest months & quarter**, **precipitation seasonality**, **precipitation of warmest and coldest quarter**, mean diurnal range, isothermality, temperature seasonality, max temperature of warmest and coldest months, temperature annual range | | | |
| *Land cover* | **Water**, bare, flooded vegetation, crops, trees, snow, grass, built-up, shrubs and scrubs | 10 | 2017 | Dynamic World |
| *Land use* | GHSL Built Surface / Volume 2015 & 2030, residential surface 2018 | 100 | 2015, 2018, 2030 | Global Human Settlements Layers |
| *Topography* | Elevation, slope | 1000 | 2000 | WorldClim |

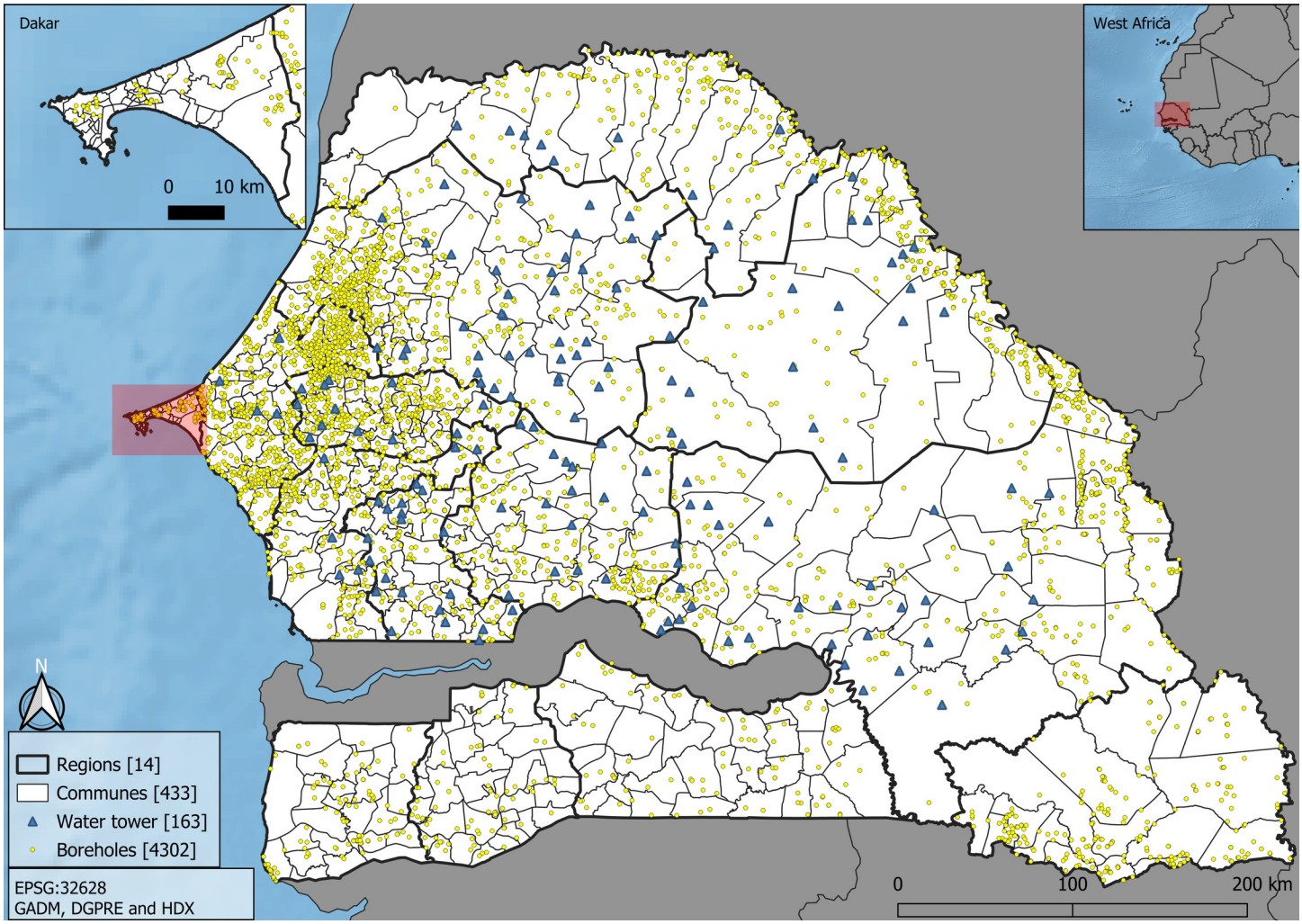

**Fig 1. Study area with water tower, borehole locations and administrative divisions in Senegal (2013).**

*Normalized difference moisture index and normalized difference water index.* The Normalized Difference Moisture Index (NDMI) and the Normalized Difference Water Index (NDWI) are directly linked to the hydrological cycle by describing soil moisture or crop water stress [30, 31]. They can be used to study water resources in an area and their impact on crops water stress.

The NDMI is an indicator of water stress within the vegetation between -1 (water stress) and 1 (no water stress). The NDWI, on the other hand, is used to determine the water content in the vegetation and ranges from -1 to 1, with values between -1 and 0 indicating no water content and values between 0 and 1 representing the percentage of water content.

The NDMI is calculated following [32]:

$$NDMI = \frac{(NIR - SWIR)}{(NIR + SWIR)}$$

and the NDWI is calculated by [33]:

$$NDWI = \frac{(GREEN - NIR)}{(GREEN + NIR)}$$

These indices are calculated using Sentinel-2 data, which has a high spatial resolution of ten metres for the visible and NIR bands and a high temporal resolution of five days since 2017. As averaging is necessary in subtropical regions to reduce the effect of cloud interferences on data acquisition [34], this was done here over 2017 as the images were acquired by two Sentinel-2 satellites working together.

*Climate & environment*: *Precipitation*. Due to the oscillation of the intertropical convergence zone, the country has two main seasons: the rainy season between June and October and the dry season for the rest of the year. The northern part of the country receives less rainfall (200 mm/year) than the south (1200 mm/year) [35]. Historically, the country has always been subject to erratic rainfall patterns on inter-annual and inter-decadal time scales, and, today, climate change is making the rainy season drier and the dry season longer [35]. As most people (70%) are employed in the agricultural sector, rainfall affect food production, jobs and therefore the location of people within the country [36].

To take into account this specific rainfall pattern over a year, several rainfall related data were extracted from [37] over the period 1970–2000: the mean annual precipitation, the precipitation of the wettest and driest month, the precipitation seasonality (ratio between the standard deviation and the mean of 12 monthly average precipitation), the precipitation of the wettest and of the driest quarter (i.e. four months), the precipitation of the warmest and coldest quarter and the mean diurnal range.

**Land cover.** *Water*. The location of water bodies is one of the key elements of human location, with more than 90% of the world's population living within 10km of water bodies [38]. We extracted the land cover class 'water' from the Dynamic World classification made with Sentinel-2 in 2017.

**Data preparation.** As all covariates have different spatial resolutions, formats and extents, a template raster grid of 1000 x 1000 metres in WGS 84 / UTM zone 28N (EPSG:32628) was used to resample all gridded covariates (e.g., NDVI, NDWI, land cover) before further processing. For land cover data, the distance to each class is calculated from a binary grid mask (1 = class considered, 0 = other classes). For vector-based datasets (e.g., borehole location, road network) the distance to each cell is calculated. All covariates are then aggregated at the administrative level by taking the weighted average of the covariate values by administrative boundary.

**Population data.** The population data used in this study came from the last available census in Senegal, conducted in 2013. We then linked the population counts from the census to a spatial administrative division of the country, obtained from the GADM. Specifically, we used the administrative level known as GADM level 4, which consists of 433 units and provides the finest spatial resolution available in the census. As the total population per administrative unit is skewed to the right and affected by unit size, log population density is used as the outcome in all our models (see S1 Fig).

Senegal is divided into two main areas: the west, which has the highest population density recorded, and the east, which is mostly rural and sparsely populated (Fig 2). Within this less populated part of the country, population densities varies widely as it includes cities such as Saint-Louis, one of the main cities in the northern part of the country and remote areas, such as in the agro-pastoral region. To consider this heterogeneity along urban-rural gradient (rural, peri-urban, ...) and understand relationship between covariates and population density

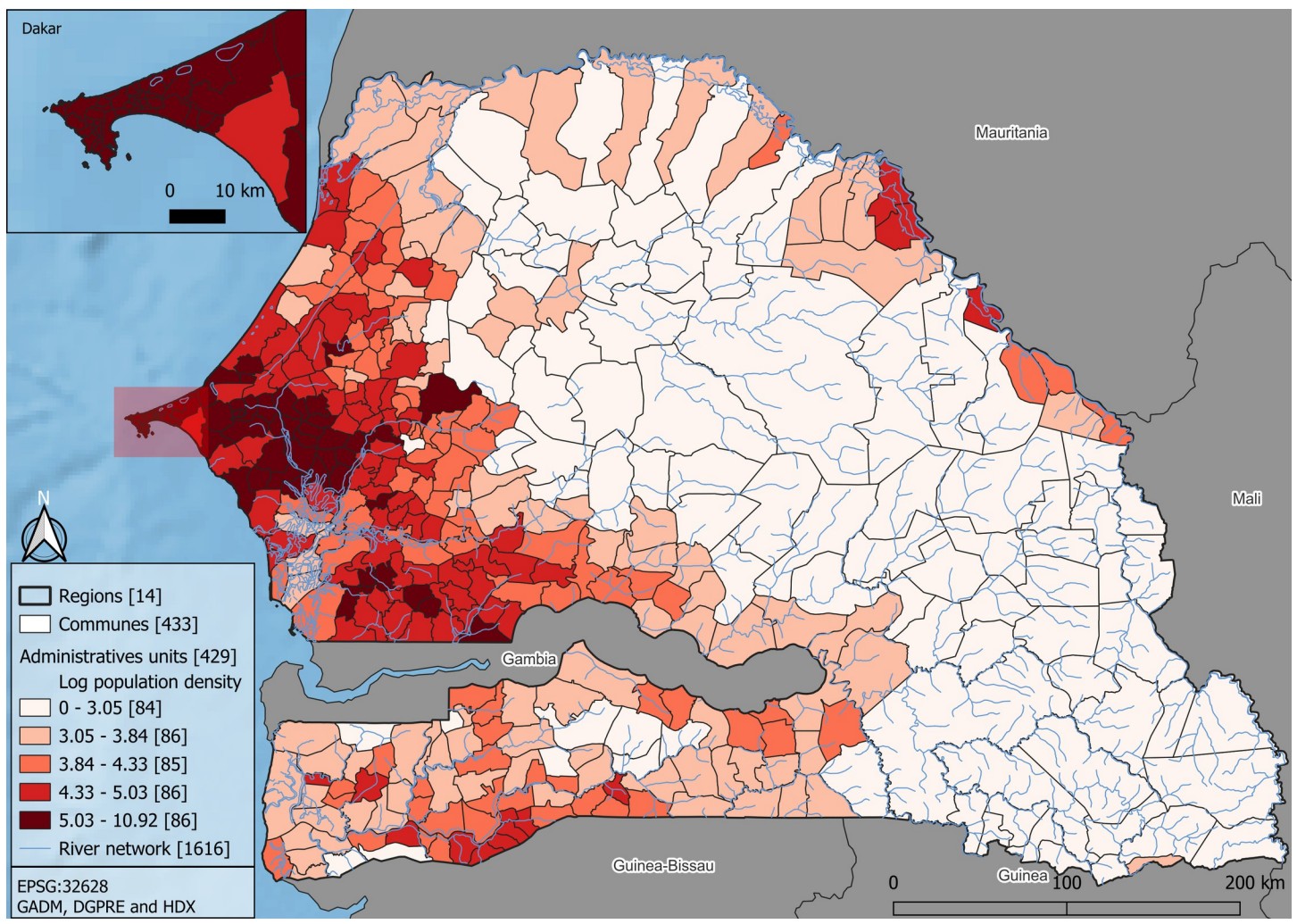

**Fig 2. Log density of population for Senegal by quantiles in 2013 and river network.** Common values in the interval are included in the upper class.

in different specific context, we divided the population density of Senegal into four sub-populations based on a quantile classification. Each of these quantiles will be part of a different model's outcomes: the first quantile is labelled as "Low Population", the first and the second quantiles will be labelled as "Low to Middle Population", "the Middle to High Population" is the first 3 quantiles and the "High Population" is the whole population of Senegal. Our modelling approach will consider each of these quantile groups as different outcomes, with the reference case being the model using all of Senegal's population densities (i.e., the "High population") as this represents the common modelling approach. In Fig 2, each quantile belongs to at least one group. This view of the population in space allows us to specify an urban-rural gradient based on the population density to better understand the link of each covariate along our population density gradient. We will then compare the relative importance of our set of covariates between each of these groups.

## Methods

To identify important covariates in modelling population density in Senegal, we use a random forest algorithm according to [11]. The full workflow is shown in Fig 3 and described in more detail in the following sections.

**Random forest.** Random forest (RF) is a machine learning algorithm used for both classification and regression tasks [39]. It is an ensemble method that combines the predictions of multiple decision trees to make more accurate predictions and avoid overfitting. Multiple decision trees are constructed using different subsets of the training data, based on a technique called 'bootstrap aggregation' or 'bagging' [39]. Each tree is constructed by randomly selecting a subset of features from the available features, leading to reduced correlation between the tree in the forest [40]. Decision trees are grown by recursively splitting the selected subset of data based on the selected features. The splits are determined by selecting the feature that optimally separates the data into homogeneous subsets. Once all the decision trees have been trained, predictions are made by each tree for the given input. The aggregation of multiple trees allows for a reduction of the variance in the output. The final prediction is the average of the predictions across all trees, making it robust to outliers. The random-forest has been shown to handle excellent performance when dealing with high dimensional dataset [40].

The trained RF model is then used to make predictions on new data by passing the input features through each decision tree in the forest and combining the predictions. RF models depend on hyperparameters such as the minimum number of observations per node (minimal node size, ranging from 1 to 10), the ratio of covariates to be used for splitting at each tree node (mtry.ratio, ranging from 10% to 100%), the fraction of observations to be used in each

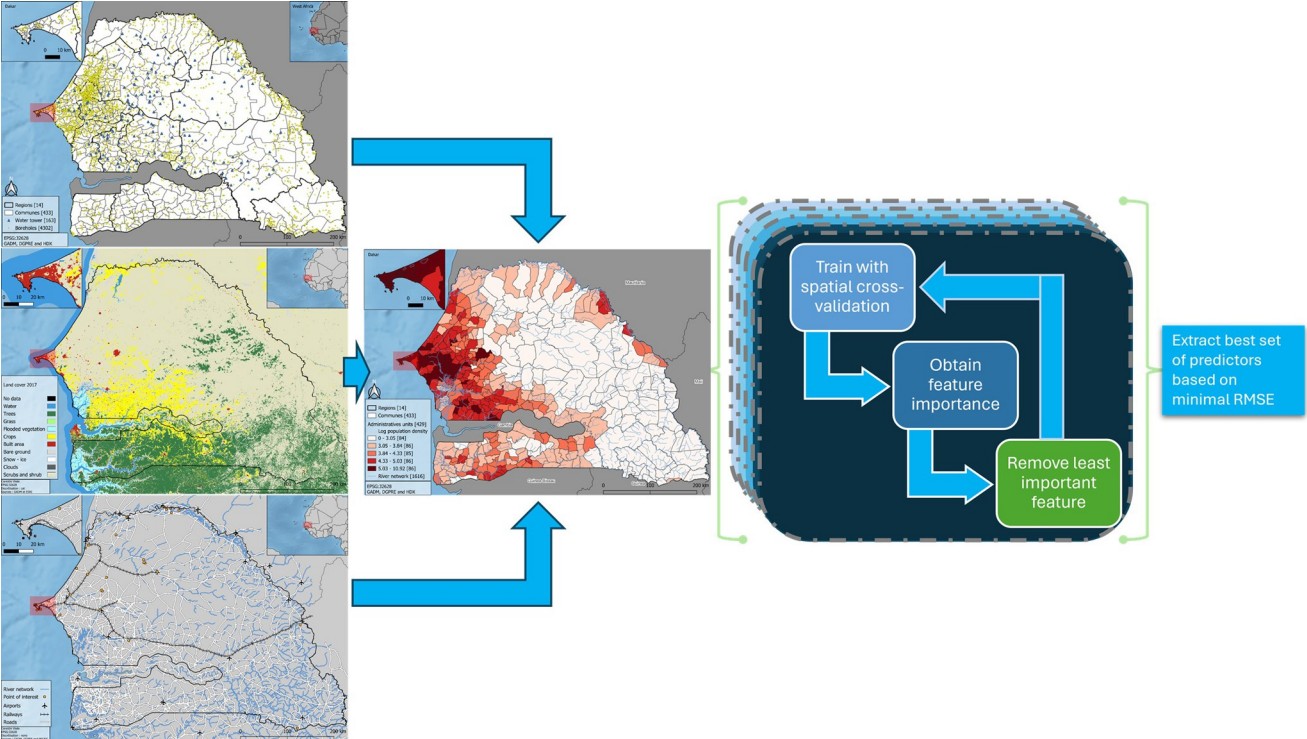

**Fig 3. Recursive feature elimination with tuning of random-forest hyperparameters in spatial cross-validation.**

tree (sample fraction, ranging from 0.1 to 1) and the number of trees to grow (ntree, fixed at 500) [39].

RF assesses the importance of features by indicating which variables have more influence on the predictions by randomly permuting the values of the covariates and assessing the gain or loss in prediction accuracy by the change in root mean square error (RMSE):

$$RMSE = \sqrt{\frac{1}{n} \sum_{i=1}^{n} \left( X_{obs,i} - X_{\text{model},i} \right)^2}$$

With n being the number of observations, $X_{obs,i}$ the observation I and $X_{model,i}$ the model output i.

**Tuning of hyper-parameters, importance evaluation and performance assessment.** RF models are optimised using three resampling loops: in the inner resampling loop, the hyper-parameters are tuned specifically for each feature subset. Then, in the middle resampling loop, the tuned hyperparameters are evaluated to determine the optimal feature subset. The outer loop is used for unbiased evaluation of the entire process [41], and we have reported model performance in terms of RMSE, $R^2$, mean absolute error (MAE) and mean absolute percentage of error (MAPE) [42]:

$$MAE = \frac{1}{n} \sum_{i=i}^{n} \left( X_{obs,i} - X_{\text{model},i} \right)$$

$$MAPE = \frac{1}{n} \sum_{i=i}^{n} \left| \frac{\left( X_{obs,i} - X_{\text{model},i} \right)}{X_{obs,i}} \right|$$

$$R^2 = 1 - \frac{\sum_{i=1}^{n} \left( X_{obs,i} - X_{\text{model},i} \right)^2}{\sum_{i=1}^{n} \left( X_{obs,i} - \overline{X_{obs,l}} \right)^2}$$

with $\overline{X_{obs,l}}$ the mean of all observations.

Covariate importance can be affected by correlation between covariates and spatial autocorrelation within covariates [43, 44]. This can lead to erroneous performance and biased variable importance [42, 45, 46]. To account for this, we built RF models with cross-validation using the spatial coordinates of the centroid of each administrative unit (i.e., spatial cross-validation).

The performance estimation level is divided into five spatial folds using the k-means algorithm, and performance estimation is repeated five times, resulting in 25 performance estimates. The hyperparameter tuning level uses 5 spatial folds (inner loop), which ultimately reduces the spatial autocorrelation bias in hyperparameter tuning [46]. Parameters are tuned by randomly searching through all combinations of mtry, node size and sample fraction with a maximum of twenty-five models to reduce overfitting.

**Recursive feature elimination.** The goal of Recursive Feature Elimination (RFE) is to select the most important covariates in a dataset [47]. It recursively eliminates features from the set of covariates to determine the most relevant subset of features by minimising the RMSE [48]. RFE helps to identify the features that contribute most to the predictive power of a model, while discarding less important, correlated and redundant features, making it a robust approach to handle multicollinearity and retaining only the most important covariates [48, 49]. To reach consensus on covariates classified as important, we build multiple (25) RFE

algorithms. This allows us to quantify the uncertainty around the selection of covariates, as they can be affected by the train-test split.

**Partial dependency of covariates.** The final RF models learned a relationship between log population density and each covariate. We calculated the partial dependence of a target variable on a covariate variable by averaging the model predictions over all possible values of that covariate in each quintile group, while holding the values of the other covariates fixed [44, 50]. It is defined as, for a regression:

$$\widehat{f}_S(x_S) = E_{X_C}\left[\widehat{f}(x_S, X_C)\right] = \int \widehat{f}(x_S, X_C) dP(X_C)$$

With $x_s$ the features we are interested to plot, $X_c$ the others features and $\widehat{f}$ our random-forest model. Our partial function $\widehat{f}_S$ is estimated by a Monte Carlo method:

$$\widehat{f}_S(x_S) = \frac{1}{n}\sum_{i=1}^{n}\widehat{f}\left(x_S, x_C^{(i)}\right)$$

With $\widehat{x_C^{(i)}}$ the values of features we are not interested in, and n is the number of instances in our dataset. This allows us to examine the marginal effect of a particular covariate on the target variable, while controlling for the influence of other covariates. This is calculated for all covariates selected in the 25 best subset of covariates for each of our quintile groups, resulting in four PD per covariate (one for low population, low to middle, middle to high and high population).

## Results

### Model performance

[Fig 4] and [Table 2] show the prediction performance ($R^2$, MAPE, MAE, RMSE) of the final RF models trained on the different population quantiles (see [S1 Table]) for hyperparameter tuning results). The middle to high population quantile achieves the highest overall score with a mean $R^2$ of 0.90 and MAPE of 0.09. The middle to high population quantile achieves a higher mean $R^2$ (0.77) and a lower mean MAPE (0.09). This quantile also has the lowest standard deviation of all quantiles in terms of both MAPE (0.04) and $R^2$ (0.12). On the other hand, the highest standard deviation is obtained by the High Population quantile, indicating highly dispersed results in the outer performance folds (all performance indicators). The low population quantile and the low to middle population quantile show comparable results, with higher dispersion for the low population quantile in all four performance indicators.

### Selected covariates

Each of the 25 RFE identified a different set of covariates by minimising the RMSE. We calculated the selection ratio of each covariate, expressed as a percentage to estimate the consensus on the importance of each covariate. [Fig 5] shows the top 10 selected covariates.

For the high population quantile, the highly selected covariates (selection ratio >75%) consist of all GHSL_* datasets (100%), distance to build-up areas (96%), nighttime light (76%), distance to boreholes (76%) and distance to financial services (76%). The first four selected covariates are related to the location of cities and built-up areas. The NDMI is the only water-related index to be selected with a ratio of 0.44 in the high population quantile.

For the other three population density quantiles, the RFE selected more often the distance to populated places (from 52 to 96%) and the distance to boreholes (from 76 to 100%). All 25 iterations of the variable selection agreed on the importance of distance to boreholes for the

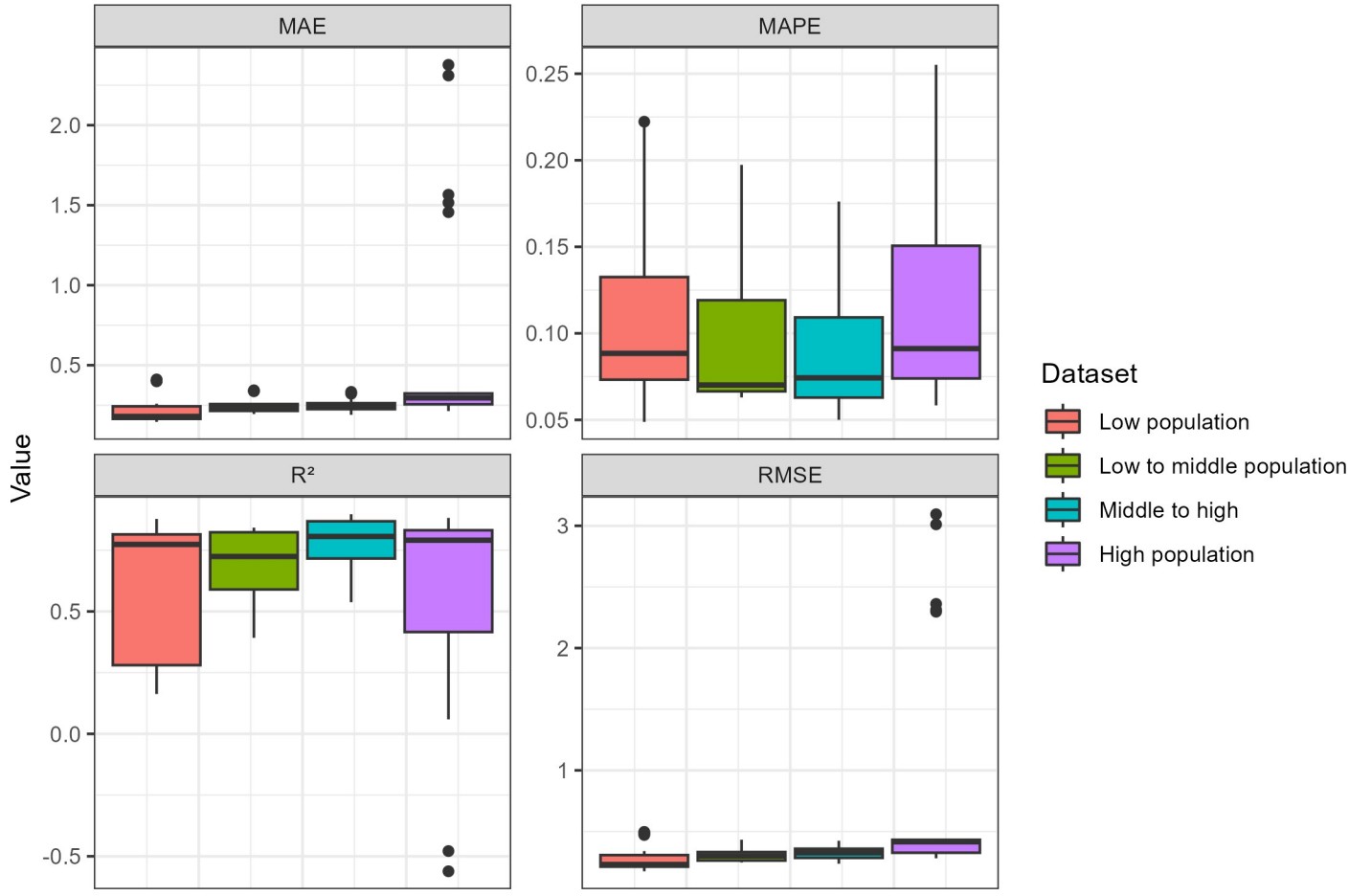

**Fig 4. Outer performance estimate between each quantile for 25 RF models.**

three quantiles. Some covariates identified as important in the high population quantile are not selected in quantile of middle to high population, such as nighttime light and distance to financial services.

In the middle to high population quantile, new covariates reach 50% of the selection rate: the Longitude and the distance to shrub and scrub (72%). Distance to crops almost reaches 50% with 48% of the selection rate. The selection rate of distance to boreholes drops from 100% to 56%, i.e., half of the RFE still select this covariate.

**Table 2. Mean with standard deviation and median performance of the 25 RFE models.**

| Densities gradient | Dataset | Mean MAE | Mean MAPE | Mean RMSE | Mean R² | Median MAE | Median MAPE | Median RMSE | Median R² |
|---|---|---|---|---|---|---|---|---|---|
| Low densities | Low population | 0.23 +- 0.09 | 0.11 +- 0.06 | 0.29 +- 0.11 | 0.6 +- 0.27 | 0.18 | 0.09 | 0.23 | 0.77 |
| | Low to middle population | 0.25 +- 0.05 | 0.1 +- 0.05 | 0.32 +- 0.06 | 0.68 +- 0.15 | 0.23 | 0.07 | 0.30 | 0.72 |
| | Middle to high population | 0.25 +- 0.04 | 0.09 +- 0.04 | 0.33 +- 0.06 | 0.77 +- 0.12 | 0.25 | 0.07 | 0.33 | 0.81 |
| High densities | High population | 0.59 +- 0.67 | 0.12 +- 0.06 | 0.82 +- 0.93 | 0.57 +- 0.42 | 0.30 | 0.09 | 0.42 | 0.79 |

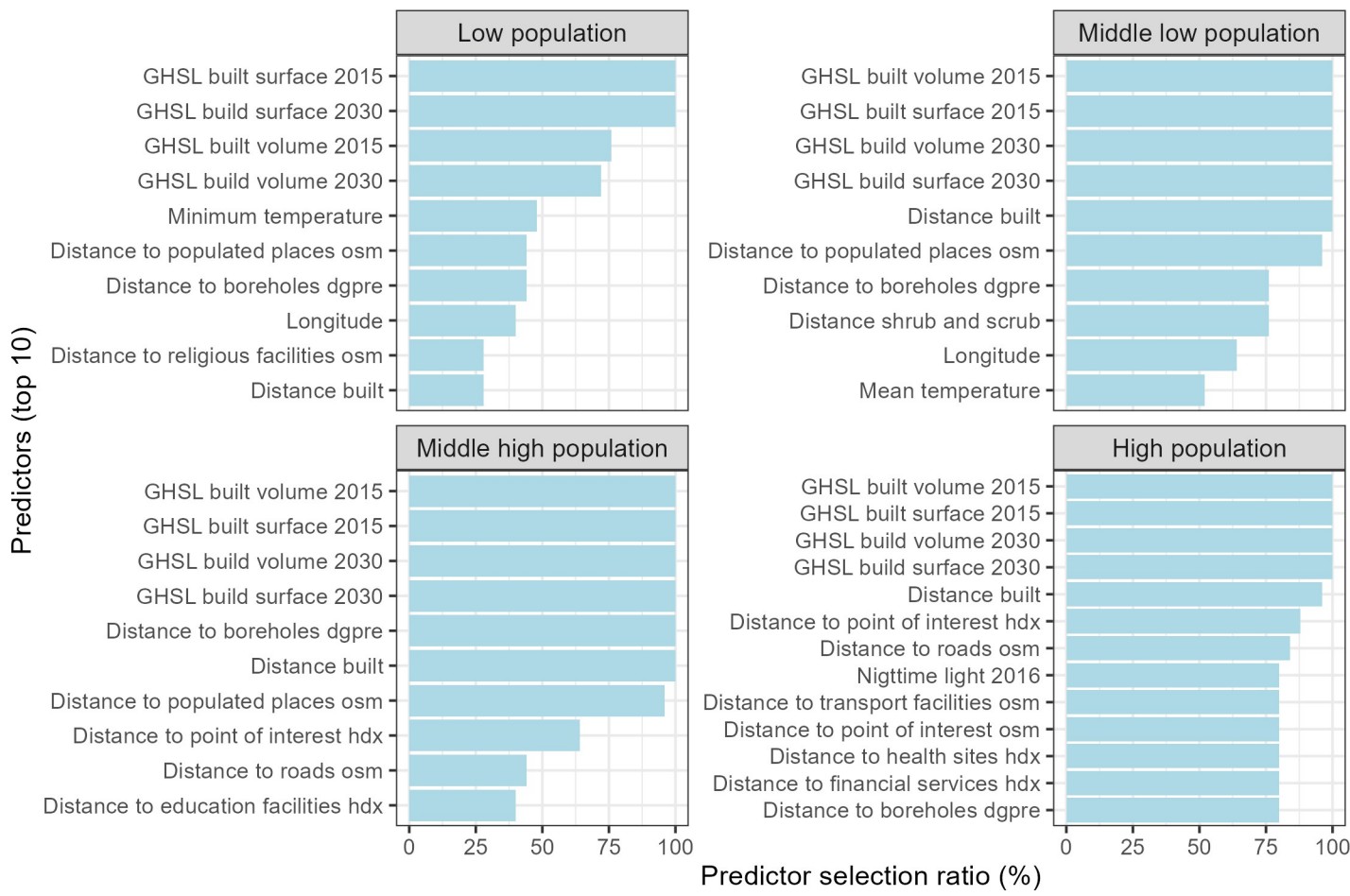

**Fig 5. Percentage of selection covariates (top 10) of log population density in each batch.**

Covariates selected in the low population quantile are more diverse, with no consensus on variable selection, with the highest selected covariate at 48%. For water-related covariates, the highest selection rate is 16% for distance to boreholes. The top 10 covariates are shown in Table 3.

**Table 3. Covariate selection ratio of RFE models.**

| Covariates | Low population | Middle to low population | Middle to high population | High population | Mean percentage |
|---|---|---|---|---|---|
| GHSL build surface 2030 | 100 | 100 | 100 | 100 | **100** |
| GHSL build surface 2015 | 100 | 100 | 100 | 100 | **100** |
| GHSL build volume 2015 | 76 | 100 | 100 | 100 | **94** |
| GHSL build volume 2030 | 72 | 100 | 100 | 100 | **93** |
| Distance to build areas | 28 | 100 | 100 | 96 | **81** |
| Distance to boreholes | 44 | 76 | 100 | 80 | **75** |
| Distance to populated places (OSM) | 44 | 96 | 96 | 60 | **74** |
| Longitude | 40 | 64 | 24 | 68 | **49** |
| Distance to points of interest (HDX) | 0 | 28 | 64 | 88 | **45** |
| Distance to roads (OSM) | 20 | 20 | 44 | 84 | **42** |

## Importance of selected covariates

The top 10 most important covariates show consistent results across quantiles (Fig 6). For quantiles 1 to 3, the top five covariates consist of GHSL_* or derived covariates. In the high population quantile, nighttime light is the most important covariate. Distance to boreholes is one of the most important covariates in each quantile, except in the high population quantile. Except for the high population quantile, all covariates have a narrow interquartile range of 0.05 to 0.95, indicating high confidence in predicting log population density.

## Partial dependency plot

Partial dependency plots (PDPs) describe the shape of the predicted relationship between the outcome variable (population density) and the covariates. By holding all the selected covariates constant except the one of interest, we can plot the average predicted log density at each quantile against the covariate of interest. We generated PDPs for each quantile for some of the new water-related covariates introduced in this study, notably distance to boreholes (Fig 7). As the variable selection algorithm is implemented 25 times, multiple curves are computed as we generate one PDP for each model selecting the covariate.

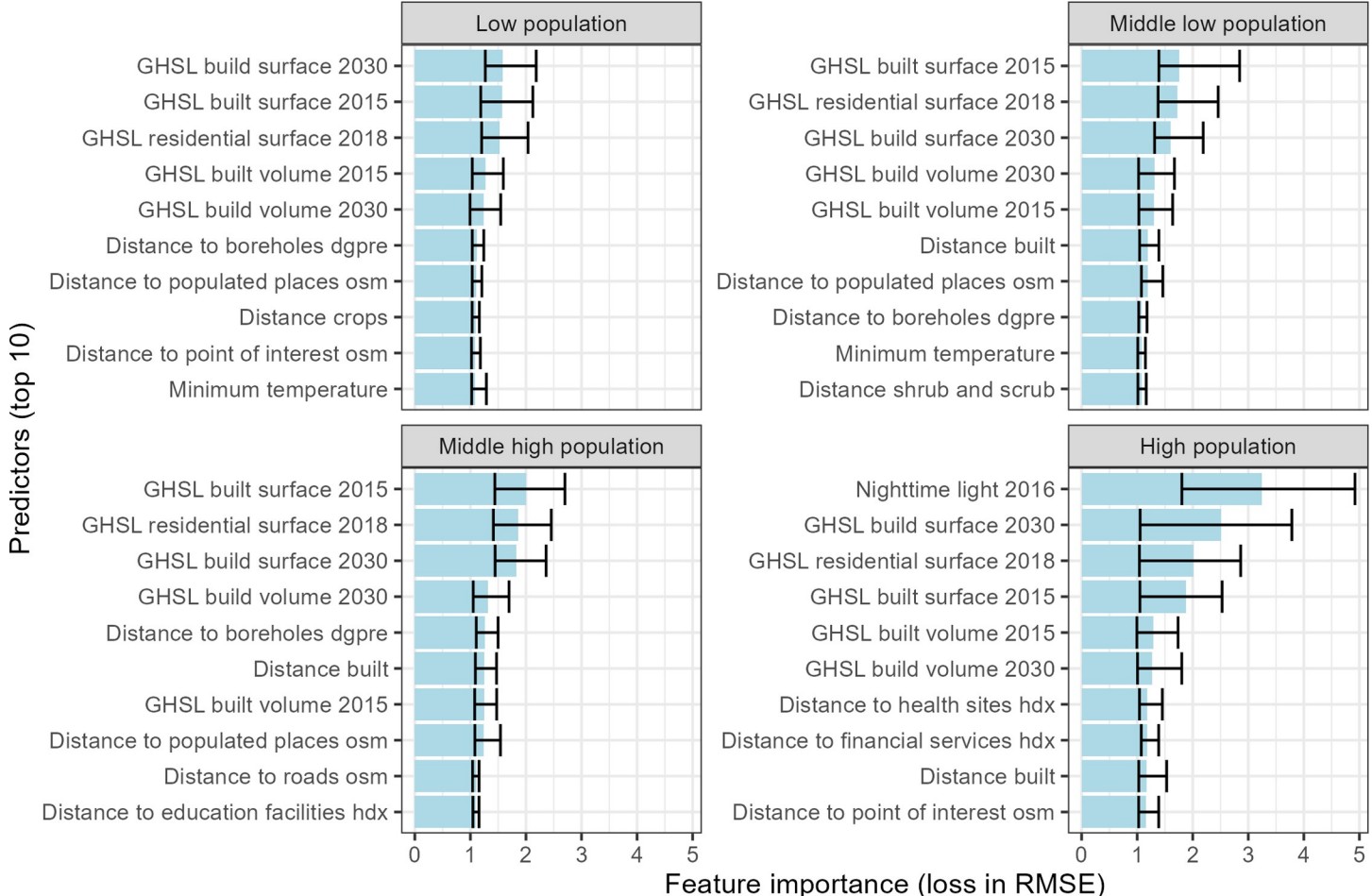

**Fig 6. Covariate importance measured as the loss in RMSE by quantile.** Blue bars show the mean importance computed from all importance when the covariate was selected. The upper and lower error bars indicate the lowest and highest importance achieved among all iterations when the covariate was selected.

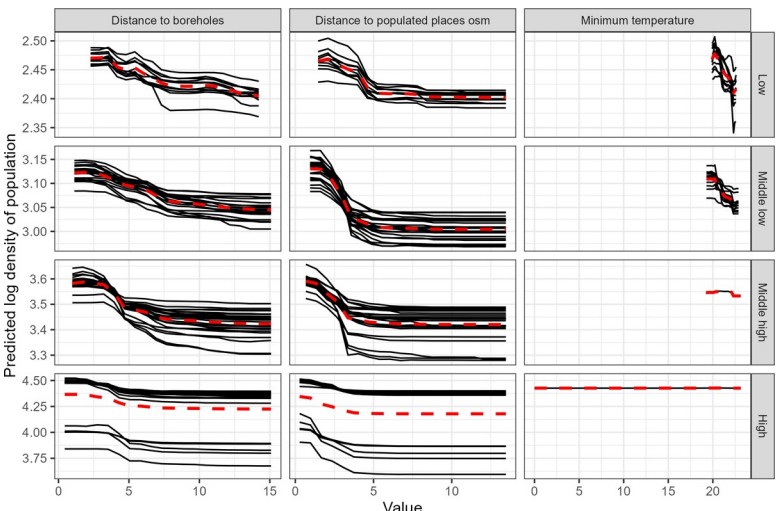

**Fig 7. Partial dependency plot for 5 covariates by quantile.** Each plain grey line represents one RF model run (over 25), while the red dotted line is the mean of estimated values over all models. The y-axis represents the population density values (in log hab/km$^2$) used in the model and the x-axis represents the value of the covariate (in kilometres for distance-related covariates and in ˚C for temperature).

For distance to boreholes, Fig 7 initially shows a slightly positive increase in population density with distance to boreholes before switching to a negative relationship, indicating that after a tipping point, the farthest distance from a borehole is associated with decreasing population density. Compared to other water-related covariates, the negative relationship is on a par with the distance to populated places from OSM. For rural areas (low population quantile), the negative relationship between distance to boreholes and distance to populated places persists at high distances (>5km), but not for other covariates after reaching a threshold, such as nighttime light (not shown) or distance to populated places.

For nighttime light, the relationship with log population density is positive up to 5 nW/cm2/sr. After this tipping point, the PDPs show flat curve with no slope. For the minimum temperature, the relationship is mostly flat for the high population quantile, meaning that there is no difference in population density regarding minimum temperature recorded in 1970–2000. In the low population quantile, the relationship is negative, with lower population density associated with higher minimum temperature.

## Discussion

The aim of this study was to understand the national-level drivers of population density by including water-related covariates and by using density batches to explicitly include urban-rural gradient. Building upon a covariate selection method and a random forest algorithm, we were able to quantify the importance of relevant drivers of population density. Our methodology allowed us to capture uncertainty in the selection of covariates by building multiple selection algorithms. Furthermore, by including water-related covariates and assessing their importance, we were able to capture their potential to better understand the distribution of population density in an urban-rural gradient compared to commonly used datasets.

Our results achieved with the whole population density show that the main covariates of population density at the national level are nighttime light and human settlements, which is in line with [13, 51, 52]. However, once high population density areas are removed from the models (i.e., when modelling low, low to middle and middle to high population), other

covariates are identified as being the most important, e.g., distance to borehole and NDMI. This suggests that the importance of covariates such as nighttime light is overestimated in national-level models and that these models may be less accurate when predicting population density outside urban centres [16]. This is consistent with [53–55], who found that nighttime light was a good proxy for the presence of urban activity and population in urban areas. Similarly, [56] found a lower correlation between nighttime light and economic activity outside urban areas. Our results therefore suggest that nighttime light is a good covariate of population density only in the most densely populated areas or when the model is built for the whole country at once.

In comparison, distance to boreholes was selected as important to understand population distribution at other levels (e.g., rural areas and other batches). This suggests that there is more information available in distance to boreholes than in nighttime light data to model rural population density in Senegal, as nighttime light is never selected in some batches. Overall, of all the water-related covariates, models using distance to boreholes, NDMI and distance to water showed high predictive performance when modelling population density in Senegal, with median $R^2$ higher than when cross-validating the model using all the country data. For the environmental covariates, only minimum and mean temperature were important. In areas of low population density, environmental covariates were selected as important covariates more often than other known covariates of population density (e.g., distance to populated places or nighttime lights).

Although distance to boreholes was an important covariate for modelling population densities, there is some variation in its selection rate (using recursive feature elimination) when high population density areas are removed from the models (i.e., when modelling low, low to medium and medium to high population densities), and it is less selected when modelling the lowest population densities (see Fig 5). These differences in selection rates may be related to the decision-making processes for water-related infrastructure development in Senegal [57, 58]. The costs associated with drilling boreholes and constructing water towers are often borne by donors and government agencies in southern countries [59]. As a result, boreholes are built where they will benefit the most people and near easy access points (i.e., maximising the population around new boreholes), i.e., in densely and moderately densely populated areas. The low selection rate of distance to boreholes for modelling the lowest population density could also be explained by socio-economic status; even when boreholes are drilled by households or by individual choice, they are financed by the richest households because they remain expensive to operate, even in cities [60]. This may explain the low selection rate in the lowest quintile and the relatively high selection rate outside the national level model.

We also assumed that all the boreholes would provide water of the same consistency and quality. However, the location of a borehole does not indicate the availability of water in that borehole. In fact, the availability of water in boreholes depends on: (i) its location in space, as some regions are more water productive than others [61]; (ii) the time of year, as there is more water available and more groundwater recharge during the rainy season; (iii) the technology used to pump water from groundwater, as hand pumps are cheaper but more difficult to operate in deep water; and (iv) the concentration of pollutants, as boreholes often draw water from shallow depths that are more susceptible to contamination [62, 63]. We have not included any of these elements here because they require additional data on seasonal availability of water at boreholes, which is difficult to collect over a long period. Furthermore, the availability of water at boreholes does not mean that the water is potable or usable for activities [64–66]. We did not include covariates of water quality because (i) it is difficult to obtain consistent data on water quality for developing countries [67], and (ii) the link between water quality and population density is more indirect, affecting health and crop productivity, which is difficult to

capture in a random forest framework [64]. In addition, although not explicitly considering water quality, health and crop productivity were included in our analysis by using vegetation indices such as NDMI, NDWI, NDVI and NPP.

In addition to identifying important drivers of population density, we assessed their relationship with population density in Senegal. Compared to environmental data and other socio-economic land uses, we found a significant negative relationship between distance to boreholes and population density, especially when densely populated areas were removed from the models. Overall, our results showed that modelling the population density at the national level is not sufficient to consider variation in the complex relationship between population density and environmental and socio-economic covariates throughout the country. We observed different relationships when working at different levels of population density (as shown on the partial dependency plots in Fig 7). Such differences may persist within the different classes of population density that were modelled in this study (i.e., low, low to middle and middle to high population densities). For example, the performance scores (i.e., $R^2$ & RMSE, MAE) had the highest standard deviation (Fig 4) when modelling the whole country at once. This could be due to the specific context of the census data for Senegal, where the distribution of values is skewed to the left, even after log-density transformation, but it could also apply to other developing countries where cities contain many more people than rural areas. The high-density class includes the capital, Dakar (1 million inhabitants for the city and 3 million for the region), but also smaller cities such as Saint-Louis (237,563 inhabitants in 2015) or Thiès (± 500,000 in 2022). The landscape is therefore heterogeneous, as is relationship between population density and socio-economic and environmental covariates.

Furthermore, the usual top-down modelling approach used in this study and in [11, 13, 68] assumes a static population. As seasonal migration is common in West Africa, the distribution of population density within countries is dynamic, and disaggregating census data on an annual basis may not be sufficient to recover the location of people [28, 69, 70]. Using the location of temporary streams and the temporal evolution of water bodies, water volumes in boreholes and satellite indices such as the NDMI, in addition to the monthly nighttime light [52], could help to provide a seasonal estimate of population density in within countries, such as done in [3, 4, 71, 72]. Regarding the temporary water bodies, they are difficult to monitor efficiently but monthly satellites estimates can be easier to use [38, 73]. Further cross-country analysis using the location/distance to boreholes as a covariate of population density, thanks to initiatives such as the Water Point Data Exchange (https://www.waterpointdata.org/) and mWater (https://www.mwater.co/), would improve knowledge of relationship between water access and population density worldwide, in addition to existing work using water-related data from the Demographic and Health Survey.

The results of this research encourage further studies to focus on cross-country comparisons within West Africa, where water-fetching is an important daily activity, and with other countries around the world to assess the potential of water-related data in the comprehension of population distribution within countries. Following other studies on access to water, [74] showed that more than half of India's population did not have access to on-site drinking water in 2011, leaving women and children to fetch water, especially in rural areas, with tube wells serving 42% of the total population in 2011 [75]. In Bolivia in 2011, DHS data showed that 54.6% and 95.6% of urban and rural households, respectively, relied on tube wells and boreholes for drinking water [76]. For Pakistan, the figures for 2012–2013 are 23% for urban households and 61.5% for rural households [77], and in Bolivia it is estimated that 71% of the total population relies on groundwater for drinking water [75]. According to these figures, fetching water from boreholes is an important part of the daily lives of rural people in regions

other than West Africa. We therefore encourage future work to replicate the methodology of this study in other regions.

The new water-related datasets used in this study have the potential to improve the prediction of rural population densities where nighttime light is absent or very limited or when the focus is only on rural areas, as to locate people affected after localised disasters. It should be noted, however, that in some contexts these data may be difficult for government agencies to collect, whereas nighttime lights are easier to monitor and more accessible for entire countries.

Including access to water could help to understanding the distribution and women separately, rather than the total population density. In West African countries, water collection is usually carried out by women and children [24, 78]. Furthermore, the relationship between people and environmental change is gender-specific [79], meaning that modelling the density of men and women as a whole could mask the importance of some environmental covariates, as was observed here with model stratification by population density class. Migration patterns are also influenced by gender, which may affect the location of men and women at the time of the census, thus confounding the importance of the covariates [80]. Future studies aimed at understanding the importance of the variable between men and women could replicate the methodology described in this paper, but instead of using the total population as the outcome, use the population density of men and women separately. This would allow the results to be compared with the current paper and to investigate gender bias in population density modelling studies.

## Conclusions

The aim of this study was to understand the main drivers of population density along an urban-rural gradient. To account for potential non-linear relationships and to control for spatial autocorrelation, we built multiple random forest models with recursive feature elimination and spatial cross-validation. By focusing on rural areas and adding water-related covariates, we found that the importance of common population density covariates is overestimated when working in rural areas. By showing that water-related characteristics (especially distance to boreholes) are important covariates of population density outside of densely populated areas, we propose a new dataset to consider in future population modelling studies, as these covariates are particularly valuable in regions where other data (e.g. nighttime light) may be lacking due to socio-economic inequalities across the urban-rural gradient within countries. Although this type of promising dataset suffers from poor global data collection, important efforts are being made by several data collectors, opening-up new opportunities for global and cross-country analysis. Furthermore, by highlighting that modelling the population density of an entire country at once leads to an overestimation of some commonly used covariates, we suggest that future studies should be conducted by splitting the population density by gender, as the relationship of males and females with the covariates may be different than when a population is modelled. Most importantly, further population mapping studies should include water-related covariates in their models to improve the estimation of population, whether it is for whole countries or rural areas. This will help to reduce urban/rural spatial inequalities and increase the inclusion of remote communities in health interventions based on population maps.

## Supporting information

**S1 Table. Results of hyperparameter tuning.**
(DOCX)

**S1 Fig.** Population metrics for Senegal in 2013, (A) is total population count per administrative unit, (B) is population density per administrative unit (#hab/km2) and (C) is the log density of population per administrative unit.
(TIF)

## Author Contributions

**Conceptualization:** Catherine Linard, Sabine Henry.

**Data curation:** Corentin Visée, Camille Morlighem.

**Formal analysis:** Corentin Visée.

**Funding acquisition:** Sabine Henry.

**Investigation:** Corentin Visée.

**Methodology:** Corentin Visée.

**Project administration:** Sabine Henry, Sébastien Dujardin.

**Resources:** Catherine Linard.

**Software:** Corentin Visée, Camille Morlighem.

**Supervision:** Sabine Henry, Sébastien Dujardin.

**Validation:** Corentin Visée, Abdoulaye Faty.

**Writing – original draft:** Corentin Visée.

**Writing – review & editing:** Corentin Visée, Camille Morlighem, Sébastien Dujardin.

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
