## [Decision Letter · Decision Letter 0]

16 Jul 2024

PONE-D-24-19766Addressing bias in national population density models: focusing on rural SenegalPLOS ONE

Dear Dr. Visée,

Thank you for submitting your manuscript to PLOS ONE. After careful consideration, we feel that it has merit but does not fully meet PLOS ONE’s publication criteria as it currently stands. Therefore, we invite you to submit a revised version of the manuscript that addresses the points raised during the review process.

We look forward to receiving your revised manuscript.

Kind regards,

Clement Ameh Yaro, Ph.D

Academic Editor

PLOS ONE

 [This work was funded by the University of Namur.].  

Reviewers' comments:

Reviewer's Responses to Questions

**Comments to the Author**

1. Is the manuscript technically sound, and do the data support the conclusions?

Reviewer #1: Partly

Reviewer #2: Yes

2. Has the statistical analysis been performed appropriately and rigorously? 

Reviewer #1: Yes

Reviewer #2: Yes

3. Have the authors made all data underlying the findings in their manuscript fully available?

Reviewer #1: Yes

Reviewer #2: Yes

4. Is the manuscript presented in an intelligible fashion and written in standard English?

Reviewer #1: Yes

Reviewer #2: Yes

5. Review Comments to the Author

Reviewer #1: The manuscript presents a thorough analysis of the relationship between population density and water-related features, particularly the distance to boreholes in Senegal. The study is methodologically sound, using random forest models to handle the complex, non-linear relationships in the data. However, there are areas where the manuscript could be enhanced for greater clarity and impact:

1. Model Specifics: Although the random forest approach is explained and the reasons for using it well established, the manuscript does not include precise information about the model's characteristics (such as the number of trees or tree depth). Transparency would increase and replication would be possible with the provision of this data.

2. Data Limitations: The study acknowledges the challenges in collecting consistent water quality data and the limitations of seasonal variations. It would be helpful to go into greater information about these restrictions, how they were managed, and how they might affect the results.

3. Exclusion of Densely Populated Areas: To concentrate on the relationship between boreholes and population density in less inhabited regions, the manuscript excludes densely populated areas. Although this is understandable, it would improve the text to examine how this exclusion might affect the overall results.

4. Partial Dependency Plots: Although partial dependency charts are a useful tool, a more thorough explanation of the data is recommended, especially regarding the various population density classes.

5. On the statistical analysis, it appears appropriate and rigorous. I think the use of random forests is well-suited to the data and research questions. The performance metrics (R², RMSE, MAE) provided are useful for evaluating model accuracy. Nonetheless, it would be helpful to have more information on the parameters used in the random forest models, and the cross-validation approach would enhance the robustness and replicability of the study. Regarding the skewness in population density values through log-density transformation, were there any validation methods taken? Lastly, is there a reason that no confidence intervals were included for key results? I think that would be helpful in contextualizing the statistical significance of the findings, but perhaps there is a reason for doing it like this.

6. Gender-Specific Analysis: The discussion on the potential for gender-specific analysis is intriguing. Including some preliminary results or a detailed plan for future gender-specific studies would add more depth to this discussion.

7. Future Research: The manuscript suggests several promising avenues for future research, such as using temporary water bodies and monthly satellite estimates. Including references demonstrating the feasibility of these approaches would be beneficial.

Reviewer #2: High resolution population density data can provide a data base for relevant research. This study focuses on the degree of precision of demographic data in rural Africa, comparing the performance of different population density products and the differences in their influencing factors. The study uses appropriate research methodology and gives credible findings. There are two main problems:

One is that the discussion and conclusion are placed together and it is recommended to separate them.

The second is that the article targets rural areas in Africa, so the influence of water availability is significant. How applicable is the study to rural areas in other continents? It is recommended to add relevant discussion.

6. PLOS authors have the option to publish the peer review history of their article (what does this mean?). If published, this will include your full peer review and any attached files.

Reviewer #1: No

Reviewer #2: No

---

## [Author Response · Author response to Decision Letter 0]

26 Aug 2024

Response to reviewers 

We would like to thank the Editor and the reviewers for their comments and thorough reviews of this manuscript. We hope to have satisfactorily addressed most of their comments in the revised version and we have provided a detailed answer to each of them.

Reviewer 1

Comment 0:

The manuscript presents a thorough analysis of the relationship between population density and water-related features, particularly the distance to boreholes in Senegal. The study is methodologically sound, using random forest models to handle the complex, non-linear relationships in the data.

Response 0:

Thank you, we are pleased to read that you find our paper a significant and impactful contribution. We hope that we have addressed your concerns in the revised version of the paper and in our point-by-point responses below.

Comment 1: 

Model Specifics: Although the random forest approach is explained and the reasons for using it well established, the manuscript does not include precise information about the model's characteristics (such as the number of trees or tree depth). Transparency would increase and replication would be possible with the provision of this data.

Response 1:

Thank you for raising this point. Model hyperparameters as number of trees or tree depth were tuned during the repeated cross-validation process, to optimize parameter values. We have added the results of the hyperparameter tuning in the supplementary material document to make it easier to reproduce the results of the paper. We have also modified the section on hyperparameter tuning to include information on the values the hyperparameters take when they are not optimised. 

Comment 2:

Data Limitations: The study acknowledges the challenges in collecting consistent water quality data and the limitations of seasonal variations. It would be helpful to go into greater information about these restrictions, how they were managed, and how they might affect the results.

Response 2:

The use of water quality data is indeed a challenging task in population modelling studies. Therefore, we decided not to include this information in our modelling. To ensure clarity on water point quality, we have added the following information in the text: (1) we have assumed that all water points have the same water quality, ensuring that all water points have the same use and productivity, and (2) we have discussed this hypothesis in the discussion section, pointing out that water quality has more of an indirect effect on population density, mostly through crop productivity, which is included in our vegetation indices. For this reason, we have not included this information in the selection of variables.

Comment 3:

Exclusion of Densely Populated Areas: To concentrate on the relationship between boreholes and population density in less inhabited regions, the manuscript excludes densely populated areas. Although this is understandable, it would improve the text to examine how this exclusion might affect the overall results.

Response 3:

Following your suggestion, we have improved the section on population density splitting in the data explanation, giving more information on the boundaries we have chosen. We have also clarified the distinction between all our population density groups in our discussion and compared it with the 'base' case with all data. 

By dividing our population density result into quintiles, we ensured that we included a model with all data and then excluded the most densely populated areas. We then compared the results of variable selection between a model with the whole population and models with a proportion of the total population.

Comment 4:

Partial Dependency Plots: Although partial dependency charts are a useful tool, a more thorough explanation of the data is recommended, especially regarding the various population density classes.

Response 4:

Following your suggestion, we have added more information in several sections of the manuscript: (1) we have rephrased the introduction to the partial dependency plot in the Results section of the manuscript, (2) we have added more information in the legend of the partial dependency plot, (3) we have clarified some sentences in the Methods section.

Comment 5:

On the statistical analysis, it appears appropriate and rigorous. I think the use of random forests is well-suited to the data and research questions. The performance metrics (R², RMSE, MAE) provided are useful for evaluating model accuracy. Nonetheless, it would be helpful to have more information on the parameters used in the random forest models, and the cross-validation approach would enhance the robustness and replicability of the study. Regarding the skewness in population density values through log-density transformation, were there any validation methods taken? 

Lastly, is there a reason that no confidence intervals were included for key results? I think that would be helpful in contextualizing the statistical significance of the findings, but perhaps there is a reason for doing it like this.

Response 5:

About the skewness of the population density values, we took the log density of the population, as other measures of population were largely skewed to the left (Figure 1). Although it is not recommended that random forest is gaussian distributed, it is recommended by (Stevens et al., 2015) to work with both density and log transformation. This allows us to reduce the effect of outliers in the count and density data, as well as controlling for the size effect of administrative units. We added the Figure 1 to our supplementary materials.

Figure 1: Population metrics for Senegal in 2013, (A) is total population count per administrative unit, (B) is population density per administrative unit (#hab/km²) and (C) is the log density of population per administrative unit.

As for the confidence intervals, we have included them in Fig 4 where they represent the range of our performance results in the validation folds. In Fig 5, we did not include confidence intervals because the figure represents the selection rate of each covariate among all our RFE iterations. In Fig 6, we have included confidence intervals by adding the lowest and highest importance achieved among all our RFE iterations when the covariate was selected to represent the extent to which importance varies. Regarding Fig 7, we included all partial dependency resulting from our final RF model and their mean.

Comment 6:

Gender-Specific Analysis: The discussion on the potential for gender-specific analysis is intriguing. Including some preliminary results or a detailed plan for future gender-specific studies would add more depth to this discussion.

Response 6:

Thank you for your suggestion, this is something we will investigate in our future work. We presented further steps in the discussion section for future studies that would take forward the work we had suggested in the end of the discussion section.

Comment 7:

Future Research: The manuscript suggests several promising avenues for future research, such as using temporary water bodies and monthly satellite estimates. Including references demonstrating the feasibility of these approaches would be beneficial.

Response 7:

To extent this part of the discussion, we added several reference to the use of monthly satellite estimates, including (Deville et al., 2014; Lai et al., 2019; Batista e Silva et al., 2020; Cheng et al., 2020). 

Reviewer 2

Comment 0:

High resolution population density data can provide a data base for relevant research. This study focuses on the degree of precision of demographic data in rural Africa, comparing the performance of different population density products and the differences in their influencing factors. The study uses appropriate research methodology and gives credible findings.

Response 0:

Thank you for your comments, we hope to have answered thoroughly each of the point you raised.

Comment 1:

One is that the discussion and conclusion are placed together, and it is recommended to separate them.

Response 1:

We modified the paper to include a conclusion section and modified discussion section accordingly. We also modified the abstract to reflect the conclusion section. 

Comment 2:

The second is that the article targets rural areas in Africa, so the influence of water availability is significant. How applicable is the study to rural areas in other continents? It is recommended to add relevant discussion.

Response 2:

Thank you for your suggestion. We have added additional information in our discussion section, in particular figures on water availability and access in several other countries thanks to the Demographic and Health Survey and other studies. These figures are given as a percentage of people with access to drinking water through wells in rural and urban areas.

Bibliography

Batista e Silva, F., Freire, S., Schiavina, M., Rosina, K., Marín-Herrera, M. A., Ziemba, L., Craglia, M., Koomen, E., and Lavalle, C.: Uncovering temporal changes in Europe’s population density patterns using a data fusion approach, Nat Commun, 11, 4631, https://doi.org/10.1038/s41467-020-18344-5, 2020.

Cheng, Z., Wang, J., and Ge, Y.: Mapping monthly population distribution and variation at 1-km resolution across China, International Journal of Geographical Information Science, 0, 1–19, https://doi.org/10.1080/13658816.2020.1854767, 2020.

Deville, P., Linard, C., Martin, S., Gilbert, M., Stevens, F. R., Gaughan, A. E., Blondel, V. D., and Tatem, A. J.: Dynamic population mapping using mobile phone data, Proc Natl Acad Sci USA, 111, 15888–15893, https://doi.org/10.1073/pnas.1408439111, 2014.

Lai, S., Erbach-Schoenberg, E. zu, Pezzulo, C., Ruktanonchai, N. W., Sorichetta, A., Steele, J., Li, T., Dooley, C. A., and Tatem, A. J.: Exploring the use of mobile phone data for national migration statistics, Palgrave Commun, 5, 1–10, https://doi.org/10.1057/s41599-019-0242-9, 2019.

Stevens, F. R., Gaughan, A. E., Linard, C., and Tatem, A. J.: Disaggregating Census Data for Population Mapping Using Random Forests with Remotely-Sensed and Ancillary Data, PLOS ONE, 10, e0107042, https://doi.org/10.1371/journal.pone.0107042, 2015.

---

## [Decision Letter · Decision Letter 1]

8 Sep 2024

Addressing bias in national population density models: focusing on rural Senegal

PONE-D-24-19766R1

Dear Dr. Visée,

We’re pleased to inform you that your manuscript has been judged scientifically suitable for publication and will be formally accepted for publication once it meets all outstanding technical requirements.

Kind regards,

Clement Ameh Yaro, Ph.D

Academic Editor

PLOS ONE

Additional Editor Comments (optional):

Reviewers' comments:

Reviewer's Responses to Questions

**Comments to the Author**

1. If the authors have adequately addressed your comments raised in a previous round of review and you feel that this manuscript is now acceptable for publication, you may indicate that here to bypass the “Comments to the Author” section, enter your conflict of interest statement in the “Confidential to Editor” section, and submit your "Accept" recommendation.

Reviewer #2: All comments have been addressed

2. Is the manuscript technically sound, and do the data support the conclusions?

Reviewer #2: Yes

3. Has the statistical analysis been performed appropriately and rigorously? 

Reviewer #2: Yes

4. Have the authors made all data underlying the findings in their manuscript fully available?

Reviewer #2: Yes

5. Is the manuscript presented in an intelligible fashion and written in standard English?

Reviewer #2: Yes

6. Review Comments to the Author

Reviewer #2: The authors have addressed all comments and I think this manuscript can be published at current form.

7. PLOS authors have the option to publish the peer review history of their article (what does this mean?). If published, this will include your full peer review and any attached files.

Reviewer #2: No

---

## [Editor Report · Acceptance letter]

30 Oct 2024

PONE-D-24-19766R1 

PLOS ONE

Dear Dr. Visée, 

I'm pleased to inform you that your manuscript has been deemed suitable for publication in PLOS ONE. Congratulations! Your manuscript is now being handed over to our production team.

Kind regards, 

on behalf of

Dr. Clement Ameh Yaro 

Academic Editor

PLOS ONE